# A Non-Invasive Deep Photoablation Technique to Inhibit DCIS Progression and Induce Antitumor Immunity

**DOI:** 10.3390/cancers14235762

**Published:** 2022-11-23

**Authors:** Kensuke Kaneko, Hiroshi Nagata, Xiao-Yi Yang, Joshua Ginzel, Zachary Hartman, Jeffrey Everitt, Philip Hughes, Timothy Haystead, Michael Morse, Herbert Kim Lyerly, Takuya Osada

**Affiliations:** 1Department of Surgery, Duke University Medical Center, 203 Research Drive, Rm 433A Box 2606, Durham, NC 27710, USA; 2Department of Cell Biology, Duke University School of Medicine, Durham, NC 27710, USA; 3Department of Pathology, Duke University School of Medicine, Durham, NC 27710, USA; 4Department of Pharmacology and Cancer Biology, Duke University School of Medicine, Durham, NC 27710, USA; 5Department of Medicine, Duke University Medical Center, Durham, NC 27710, USA

**Keywords:** photodynamic therapy, heat shock protein 90, anti-PD-L1 antibody, ductal carcinoma in situ

## Abstract

**Simple Summary:**

Fewer than half of ductal carcinoma in situ (DCIS) cases progress to invasive breast cancer (BC); however, most are treated surgically, which raises a concern of overtreatment. Novel approaches to the management of DCIS that do not rely on surgery are warranted. In this study, we assessed the efficacy of heat shock protein 90 (Hsp90)-targeted photodynamic therapy (HS201-PDT) as a novel alternative treatment to target DCIS and prevent progression to invasive BC. The preventive application of HS201-PDT to the mammary glands of spontaneous murine DCIS models significantly improved disease-free survival. Enhancement of immune response with systemically administered anti-PD-L1 antibodies along with repeated HS201-PDT to mammary glands was effective even in a more aggressive DCIS model. Thus, HS201-PDT monotherapy or the combination with anti-PD-L1 antibodies is a promising strategy that could serve as an alternative treatment of DCIS to prevent progression to invasive BC.

**Abstract:**

Ductal carcinoma in situ (DCIS) of the breast is often managed by lumpectomy and radiation or mastectomy, despite its indolent features. Effective non-invasive treatment strategies could reduce the morbidity of DCIS treatment. We have exploited the high heat shock protein 90 (HSP90) activity in premalignant and malignant breast disease to non-invasively detect and selectively ablate tumors using photodynamic therapy (PDT). PDT with the HSP90-targeting photosensitizer, HS201, can not only ablate invasive breast cancers (BCs) while sparing non-tumor tissue, but also induce antitumor immunity. We hypothesized that HS201-PDT would both non-invasively ablate DCIS and prevent progression to invasive BC. We tested in vitro selective uptake and photosensitivity of HS201 in DCIS cell lines compared to the non-selective parental verteporfin, and assessed in vivo antitumor efficacy in mammary fat pad and intraductal implantation models. Selective uptake of HS201 enabled treatment of intraductal lesions while minimizing toxicity to non-tumor tissue. The in vivo activity of HS201-PDT was also tested in female MMTV-neu mice prior to the development of spontaneous invasive BC. Mice aged 5 months were administered HS201, and their mammary glands were exposed to laser light. HS201-PDT delayed the emergence of invasive BC, significantly prolonged disease-free survival (DFS) (*p* = 0.0328) and tended to improve overall survival compared to the no-treatment control (*p* = 0.0872). Systemic administration of anti-PD-L1 was combined with HS201-PDT and was tested in a more aggressive spontaneous tumor model, HER2delta16 transgenic mice. A single PDT dose combined with anti-PD-L1 improved DFS compared to the no-treatment control, which was significantly improved with repetitive HS201-PDT given with anti-PD-L1 (*p* = 0.0319). In conclusion, a non-invasive, skin- and tissue-sparing PDT strategy in combination with anti-PD-L1 antibodies effectively prevented malignant progression of DCIS to invasive BC. This non-invasive treatment strategy of DCIS may be safe and effective, while providing an option to reduce the morbidity of current conventional treatment for patients with DCIS. Clinical testing of HS201 is currently underway.

## 1. Introduction

Ductal carcinoma in situ (DCIS) is considered a non-obligate precursor of invasive breast cancer (BC), and is typically treated with surgery, radiation and hormonal therapy [1]. However, because of the low rate (20–30%) of progression to invasive BC reported in natural history studies, a large percentage of women may be overtreated [2,3,4,5]. To reduce the chance of overtreatment, it is imperative to develop a robust method of stratifying the patients that have aggressive DCIS from those whose DCIS is indolent. Unfortunately, however, there are no reliable markers or methods identified thus far [6,7]. Even if such markers are identified, there will still be a need for novel alternative treatments to target DCIS, if less invasive and less toxic. Our recent study on photodynamic therapy (PDT), as a local therapy for cancer, demonstrated the induction of systemic antitumor immunity against invasive BC in mice (ref manuscript submitted). PDT combines a photosensitizer (PS) compound administered systemically and tumor-targeted laser light. Laser ablation results in generation of reactive oxygen species (ROS), which leads to cellular damage [8,9]. However, most PSs currently available have only limited cancer selectivity, and thus damage to adjacent normal tissues by laser irradiation is inevitable [10,11]. Aiming at more efficient delivery and prolonged retention of the PS at the tumor site, we developed a novel Hsp90-targeting PS, HS201, which consists of verteporfin (VP, photosensitizer) tethered to an Hsp90 small-molecule inhibitor [12,13]. As Hsp90 overexpression is observed in various types of cancers, including BC, and cell surface expression of Hsp90 has been confirmed [14,15,16], Hsp90 may be a promising target molecule for drug delivery. Using human BC xenograft models and murine syngeneic BC models, we recently confirmed stronger accumulation and longer retention of HS201 in BC of various molecular subtypes compared to parental VP, and a stronger antitumor effect of PDT using HS201 (HS201-PDT) [12]. Previous studies reported the overexpression of Hsp90 by DCIS [17,18]. Therefore, we propose to study whether HS201-PDT could prevent the transition of DCIS to invasive BC.

Recent progress in cancer therapy is largely due to the development of immune checkpoint inhibitors (ICIs) targeting the PD-1/PD-L1 interaction. For the treatment of clinical BCs, FDA approved the anti-PD-1 antibody pembrolizumab in combination with chemotherapy to treat PD-L1-positive, unresectable locally advanced or metastatic triple-negative BC (TNBC) and also for the treatment of patients with high-risk early-stage TNBC in combination with chemotherapy as neoadjuvant treatment [19,20]. Other ICIs targeting PD-1 or PD-L1 are in development [21]. In our previous studies on local therapies, HIFU and HS201-PDT, against BCs, we tested the combination strategy with PD-1/PD-L1 blockade and demonstrated that these local treatments combined with anti-PD-L1 antibodies could enhance induction of tumor-antigen-specific immune response and elicit strong antitumor activity, which suppressed the growth of distant tumors as well as directly treated tumors [13,22]. Importantly, recent studies on gene expression analysis of DCIS revealed the gene mutational patterns in DCIS to be almost identical to those in invasive BC, with high-frequency mutations in PI3K and p53 genes [23,24,25]. Therefore, DCIS is expected to express tumor-specific antigens that are common with invasive BC and may function as good targets for immune-based elimination [26]. Thus, combining ICI with HS201-PDT, which not only destroys tumors by generating ROS but also induces tumor-antigen-specific immune response, may elicit strong antitumor immunity and efficacy against DCIS. In this study, we investigated whether HS201-PDT monotherapy or the combination with anti-PD-L1 can inhibit DCIS progression to invasive BC using multiple DCIS models, including human DCIS orthotopic xenograft models and spontaneous BC models in transgenic mice. We revealed that preventive HS201-PDT with/without ICI is effective at reducing the number and delaying the onset of invasive BC, and prolonged disease-free survival.

## 2. Materials and Methods

### 2.1. Cell Culture

Human DCIS cell lines, MCF10DCIS.com and SUM225CWN, were purchased from Asterand (Detroit, MI, USA). The MCF10DCIS.com cell line, which was derived from the non-cancerous MCF10A cell line, is known to form DCIS-like lesions after flank or intraductal implantation into immunodeficient mice and spontaneously progresses to invasive BC [27]. The SUM225CWN cell line was established from a chest wall recurrence in a patient treated for DCIS, and is known to form lesions similar to human DCIS after intraductal implantation into mouse mammary ducts [28,29]. MCF10DCIS.com cells were cultured in DMEM/F12 medium (Cat# 11320033, ThermoFisher Scientific, Waltham, MA, USA) supplemented with 5% horse serum (Cat# 26050088, ThermoFisher Scientific) and 2 mM glutamine (Cat# G3126, Sigma-Aldrich, St. Louis, MO, USA). SUM225CWN cells were cultured in F12 medium (Cat# 11765054, ThermoFisher Scientific) supplemented with 5% FBS (Cat# 10082147, ThermoFisher Scientific), HEPES (10 mM, Cat# H3375, Sigma-Aldrich), insulin (5 µg/mL, Cat# 19278, Sigma-Aldrich) and hydrocortisone (1 µg/mL, Cat# H0888, Sigma-Aldrich). Cell lines were screened for rodent pathogens at Duke Division of Laboratory Animal Resources (DLAR) Veterinary Diagnostic Laboratory prior to use in animal experiments.

### 2.2. Reagents and Antibodies

Verteporfin (VP, Visudyne, Novartis Pharmaceuticals Corp., Basel, Switzerland), a well-characterized PS also serving as an nIR dye with a 700 nm emission peak, and HS201, a novel PS made of VP tethered to an Hsp90 small-molecule inhibitor (HS10), were used for in vitro and in vivo imaging and PDT. The HS201 compound was developed and supplied by the Haystead Lab (Duke University Department of Pharmacology and Cancer Biology) as previously reported [12,13]. Anti-Hsp90 antibody (clone AC88, Cat# ab13492,) was purchased from Abcam plc. (Cambridge, UK). Secondary antibodies (goat anti-mouse IgG, goat anti-rabbit IgG and donkey anti-mouse IgG antibodies) conjugated with nIR dye, IRDye 800CW (Cat# 926-32210, 926-32211 and 926-32212, respectively), were purchased from LI-COR Inc. (Lincoln, NE, USA). Anti-murine PD-L1 antibody (clone; 10F.9G2, Cat# BE0101) was purchased from Bio X cell (Lebanon, NH, USA) and used for animal treatment.

### 2.3. Mice

Inbred SCID-beige mice and MMTV-neu mice were purchased from Taconic Biosciences, Inc. (New York, NY, USA) and bred at the Duke University Cancer Center Isolation Facility. MMTV-neu mice are known to develop intraductal hyperplasias that share common cytological features with human DCIS and progress to invasive BCs [30]. The HER2delta16 (HER2d16) transgenic mouse model was previously generated in our lab [31,32]. We confirmed the formation of HER2d16-driven spontaneous breast tumors in HER2d16 Tg mice once they were put on a doxycycline-containing diet (200 mg/kg, Bio-Serv, Flemington, NJ, USA) for at least 4 weeks. All animal studies described were approved by the Duke University Medical Center Institutional Animal Care & Use Committee and the US Army Medical Research and Materiel Command (USAMRMC) Animal Care and Use Review Office (ACURO) and performed in accordance with guidelines published by the Commission on Life Sciences of the National Research Council.

### 2.4. Confocal Microscopy of Cell Surface Hsp90 Expression In Vitro

MCF10DCIS.com cells were resuspended in DMEM/F12 medium supplemented with 5% horse serum, seeded on glass-bottom dishes (MatTek, Ashland, MA, USA) and incubated overnight at 37 °C. Cells were washed with PBS twice and incubated with anti-Hsp90 mAb (AC88, 1 µg/mL, Abcam) for 30 min on ice. Then, cells were washed with PBS and fixed with 10% neutral buffered formalin for 15 min at 37 °C. After the removal of formalin, the cells were then incubated with secondary antibody (1:500 dilution, AF568-conjugated goat anti-mouse IgG, Cat# A-11004, Life Technologies, Carlsbad, CA, USA), followed by Wheat Germ Agglutinin (WGA) Alexa Fluor 488 conjugate membrane staining dye (5 µg/mL, Cat# W11261, ThermoFisher) for 20 min and DAPI (Cat# 422801, BioLegend, San Diego, CA, USA) for 10 min at room temperature. After being washed with PBS, cells in the dish were observed using a ZEISS LSM880 confocal microscope (Carl Zeiss AG., Oberkochen, Germany).

### 2.5. Measurement of Cellular Uptake of Photosensitizer In Vitro

The in vitro uptake of reagents designed for nIR imaging and PDT was measured using the Odyssey CLx imaging system (LI-COR) at 700 nm wavelength. DCIS cells were seeded in 96-well plates (10,000 cells/well) and cultured in each appropriate medium at 37 °C. After the cells reached 70–80% confluency in each well, VP or HS201 (0.03–3 μM, respectively) was added and incubated for 30 min at 37 °C, and then the PS was removed, and washed twice with PBS. The signal intensity of each well was measured at 700 nm by LI-COR Odyssey imager to evaluate cellular uptake of VP and HS201.

### 2.6. Killing Assay

The effect of in vitro PDT using VP (VP-PDT) and HS201 (HS201-PDT) was determined by MTT assay. Human DCIS cells seeded in a 96-well plate were co-incubated with PS and washed as described above. A laser with a wavelength of 690 nm was applied at various doses (1.88–120 J/cm^2^) to each well using the medical laser system ML7710 and illumination system ML8500 (Modulight, Inc., Tampere, Finland). Cells were incubated overnight prior to the analysis. MTT assay was performed as previously reported [12], and the optical density value at 550 nm (650 nm subtracted) of each well was measured by a Model 680 Microplate Reader (Bio-Rad Laboratories, Inc., Hercules, CA, USA).

### 2.7. In Vivo Imaging of DCIS and Advanced BC with HS201

For the human DCIS xenograft model, MCF10DCIS.com cells (1 × 10^6^ cells/mouse) were injected into the 4th mammary fat pad of female SCID-beige mice. We also used a spontaneous BC model, MMTV-neu transgenic mice for HS201 imaging. MCF10DCIS.com tumor-bearing SCID-beige mice or spontaneous tumor-bearing MMTV-neu mice were administered 25 nmol of HS201 via tail vein, and nIR signal was imaged 6 h later using the Pearl Trilogy imager (LI-COR) at 700 nm.

### 2.8. Pathology Evaluation

Sections of selected full-thickness mammary fat pad tissues containing skin and adnexa, as well as mammary masses, were fixed in 10% neutral buffered formalin, processed routinely to paraffin, sectioned at 5 μm and stained with hematoxylin and eosin (H&E). They were evaluated by a board-certified veterinary pathologist with experience in mouse and toxicologic pathology (JE).

### 2.9. Photodynamic Therapy

Human DCIS cell lines, MCF10DCIS.com and SUM225CWN cells (1 × 10^6^ cells/mouse), were inoculated into the 4th mammary fat pad of female SCID-beige mice. After the tumor size reached 5 mm in diameter, PDT was performed. Briefly, mice received intravenous injection of HS201 (25 nmol/mouse) via tail vein injection, and 6 h later (drug–light interval (DLI) = 6 h), the 4th mammary gland was irradiated with a 690 nm laser at the doses of 30 and 120 J/cm^2^ using the medical laser system ML7710 (Modulight). These doses were selected based on our previous studies [12,13]. In the intraductal implantation model, twenty-one days after intraductal injection of MCF10DCIS.com cells (5 × 10^4^ cells in 10 µL) to the 4th mammary gland, mice received intravenous injection of HS201 (25 nmol/mouse) and laser irradiation as described above. Tumor size was monitored by caliper measurement twice a week. Tumor volume was calculated as follows: tumor volume (mm^3^) = short diameter (mm) × short diameter (mm) × long diameter (mm)/2. Mice were euthanized when tumor volume reached humane endpoint according to IACUC-approved protocol (>2000 mm^3^).

In the MMTV-neu DCIS model, female mice aged 151–164 days were shaved to expose the skin above all the mammary glands. HS201 (25 nmol/mouse) was intravenously administered via tail vein injection, and 6 h later, 690 nm laser irradiation (30 J/cm^2^) was repeated for the 7 areas that included all mammary glands (Appendix A) under general anesthesia. The lower laser dose (30 J/cm^2^) was chosen to ensure the safety of mice receiving multiple laser exposures. MMTV-neu mice were monitored weekly by palpation for the development of tumors, and tumor sizes were measured with a caliper every week. Mice were euthanized once the tumor sizes reached humane endpoints or at 85 weeks after birth if they had not reached humane endpoints.

In the HER2d16 mouse model, female mice were given a doxycycline diet starting on day 0, and HS201-PDT was performed on day 14 for all the mammary glands in MMTV-neu mice. For the combination treatment with a single HS201-PDT treatment, the anti-PD-L1 antibody (100 µg/inj) was intraperitoneally administered to mice on days 14, 17, 21 and 24. For the repeated HS201-PDT treatment with anti-PD-L1, mice received HS201-PDT once a week 4 times (days 14, 21, 28 and 35), and anti-PD-L1 (100 µg/inj) was administered to mice on the same days with PDT procedures. Tumor emergence and growth were monitored twice a week until the tumor size reached humane endpoints.

As we demonstrated that control cohorts of (i) HS201 alone without laser irradiation and (ii) laser irradiation alone without HS201 exerted no antitumor effect in our previous studies [12,13], and it was important to reduce the number of animals used in the study according to recommendation of Duke IACUC, we did not repeat these controls in the present experiments.

### 2.10. Statistical Analysis

For the evaluation of tumor volume over time, areas under tumor growth curve were calculated and compared between groups using one-way ANOVA with Tukey’s multiple comparisons test using Prism software (Graphpad, San Diego, CA, USA). For the analyses of mouse disease-free survival and overall survival, Kaplan–Meier survival curves for tumor-bearing mice were generated and log-rank tests were performed using GraphPad Prism 9.0. Multiple comparisons of survival curves were performed using a computing method installed in Prism 9.0 that matches SPSS and SAS. All tests of hypotheses were two-sided, and a significance level of 0.05 was used.

## 3. Results

### 3.1. In Vitro Uptake of HS201 by Human DCIS Cell Lines and Killing by Laser Exposure

Prior to performing in vivo prevention studies, we assessed cell surface Hsp90 expression, uptake of the Hsp90-specific, VP-conjugated PS HS201 and in vitro killing by PDT of MCF10DCIS.com, a widely used human DCIS cell line. Fluorescence microscopy confirmed clear expression of Hsp90 on the cell surface, as we previously observed in established invasive BC cell lines (Figure 1A) [12,14]. Further, uptake of HS201 by MCF10 DCIS.com and SUM225CWN, another human DCIS cell line, exceeded that of VP (Figure 1B). PDT experiments demonstrated that in both DCIS cell lines, HS201 induced stronger cytotoxicity compared to VP, demonstrating efficient killing of DCIS cells by Hsp90-targeted HS201-PDT. MCF10DCIS.com cells were relatively more sensitive to in vitro PDT compared to SUM225CWN cells, such that lower laser doses (15 or 30 J/cm^2^) were sufficient to kill MCF10DCIS.com cells (Figure 1C).

### 3.2. Tumor Growth Suppression by HS201 Photodynamic Therapy in a Mammary Fat Pad Implantation Model

We tested the antitumor efficacy of HS201-PDT against MCF10DCIS.com and SUM225CWN cells 7 days after they were implanted into the fourth mammary fat pad of female SCID-beige mice. As shown in Figure 2A(i) (red arrow), there was accumulation of HS201 at the fourth mammary fat pad after systemic administration of HS201. At both laser doses, 30 and 120 J/cm^2^, HS201-PDT modestly suppressed MCF10DCIS.com tumor growth (Figure 2A(ii),(iii)), but significantly suppressed SUM225CWN tumor growth (Figure 2B). Thus, we confirmed the antitumor efficacy of HS201-PDT in vivo using mammary fat pad implantation models of human DCIS cell lines, but the in vivo antitumor efficacy did not necessarily correspond to the killing efficacy observed in in vitro PDT.

### 3.3. Tumor Growth Suppression by HS201-PDT in Intraductal DCIS Cell Implantation Model

Because DCIS originates in mammary ducts, we wished to test HS201-PDT in a more physiologic model of DCIS, intraductal implantation of DCIS tumor cells in SCID-beige mice. In vivo tumor growth was suppressed by PDT at both laser doses (Figure 2C). Further, untreated mice showed earlier onset of tumor growth in the implanted mammary gland, while mice treated with HS201-PDT showed numerically delayed tumor incidence (Figure 2D). Incidence of tumor development was significantly lower in HS201-PDT-treated mice at the 120 J/cm^2^ laser dose (37.5%), while HS201-PDT at the 30 J/cm^2^ laser dose showed 50% incidence, and the no-treatment control showed 85.7% incidence of tumor development after intraductal implantation (Figure 2D).

Overall survival of MCF10DCIS.com-implanted mice was monitored and is shown in Figure 2E. A log-rank test was performed, which showed *p* = 0.0577 for the comparison of three groups. Mice treated with HS201-PDT at a 120 J/cm^2^ laser dose showed significantly improved overall survival when compared to mice in the no-treatment group (*p* = 0.0139). Median survival was 72 days for the no-treatment group, 130 days for the HS201-PDT (30 J/cm^2^)-treated group and 328 days for the HS201-PDT (120 J/cm^2^)-treated group.

These data suggest that higher doses of laser energy result in greater survival.

### 3.4. Preventive Application of HS201-PDT in MMTV-Neu Mice at the Age of 5 Months Induced Inflammatory Response in Breast Tissues

Because we confirmed growth suppression of implanted human DCIS cell lines in vivo by HS201-PDT, we sought to test the effect of HS201-PDT in spontaneous DCIS models using transgenic mice. We used MMTV-neu mice, which develop rat-neu-driven spontaneous BC at the median age of 11 months, but can also develop it as early as 7 months (author observation) (Appendix A). We confirmed that HS201 accumulated in the early breast lesions of these mice (Figure 3A), although the accumulated nIR signal was weaker compared to advanced BC in MMTV-neu mice (Figure 3A, bottom). Therefore, we decided to conduct preventive HS201-PDT on all the mammary glands at 5 months of age (151–164 days after birth). Because HS201-PDT at a 30 J/cm^2^ laser dose induced tumor growth suppression of implanted DCIS tumors similarly to HS201-PDT at 120 J/cm^2^, and because laser irradiation at 120 J/cm^2^ of the area including the right #2 and #3 mammary glands could induce mild liver damage, unlike the lower dose of 30 J/cm^2^, we chose 30 J/cm^2^ as the laser dose for preventive HS201-PDT (Appendix A).

To characterize histopathological effects induced by HS201-PDT, H&E-stained sections of PDT-treated mammary gland tissues were assessed 3 days following treatment. As a positive control, large advanced mammary masses in MMTV-neu mice were treated with HS201-PDT (120 J/cm^2^), and collected tumors were assessed for microscopic changes. The histopathologic appearance of these large treated mammary tumors was characterized by extensive hemorrhage and necrosis (Figure 3B(i)) as well as aggregates of lymphoid cell infiltrate and scattered polymorphonuclear leukocytes (PMNs) (Figure 3B(ii)). Microscopic examination of the mammary region of 5-month-old MMTV-neu mice showed early preneoplastic changes including atypical dysplastic mammary ducts (Figure 3C(i)) and proliferative focal dysplasias, known in mouse pathology as mammary intraepithelial neoplasia (MIN) (Figure 3C(ii)). These MIN lesions are considered to be the murine equivalent of human DCIS and in several sections were noted to be arising from the mammary duct epithelium (Figure 3C(iii)). Duct ectasia surrounded some of the larger MIN lesions, suggesting ductal blockage by the proliferative dysplasia (Figure 3C(iv)). By 11 months of age, many mice showed invasive BC sharing cellular features with larger MIN lesions. When compared to untreated mammary glands (Figure 3D(i)), HS201-PDT (30 J/cm^2^) treatment of 5-month-old mice resulted in inflammation associated with mammary ducts, manifesting as inflammation in the periductal and surrounding periductal adipose tissue (Figure 3D(ii)). These inflammatory changes included leukocyte infiltration of duct adventitia and focal PMN aggregates in periductal fat pad regions. Thus, with our observational pathological analysis, we confirmed that preventive HS201-PDT has a recognizable effect on the mammary ducts of 5-month-old MMTV-neu mice when early focal dysplastic lesions are present.

### 3.5. HS201-PDT Delayed the Emergence of Invasive Breast Cancer and Improved Disease-Free Survival in MMTV-Neu DCIS Model

In order to determine whether HS201-PDT could prevent progression of DCIS to invasive BC, we applied HS201-PDT to the mammary areas of 5-month-old mice, prior to predicted development of invasive BC (Figure 4A). The onset of tumor development was delayed in the HS201-PDT group compared to the no-treatment group (Figure 4B). Importantly, HS201-PDT resulted in a significant improvement in DFS (Figure 4C). Median DFS for the HS201-PDT group was 62 weeks, while that for the no-treatment group was 53 weeks. Although not statistically significant, there was a trend of longer OS for mice treated with HS201-PDT compared to the no-treatment control (*p* = 0.0872) (Figure 4D). Median OS for the HS201-PDT group was 74 weeks and 64 weeks for the no-treatment group.

To determine whether earlier treatment was more effective, we also tested late (8 months following birth) HS201-PDT treatment for MMTV-neu mice (Figure 4E). Here, the impact of HS201-PDT on DFS was less than in the earlier treatment. Although there was a trend, there was not a statistically different DFS and OS in the HS201-PDT-treated group compared to the no-treatment control (*p* = 0.194 for DFS, *p* = 0.0683 for OS) (Figure 4F,G). Median DFS and median OS for the late HS201-PDT treatment group were 64.57 and 80.14 weeks, respectively. Based on these findings, we confirmed that preventive HS201-PDT conducted at 5 months of age for MMTV-neu mice effectively delayed and reduced invasive tumor occurrence in mice, and significantly improved their DFS. HS201-PDT performed at a later time point (8 months of age) could also delay the tumor emergence, with a trend of improved OS but no significant differences in DFS compared to the no-treatment control.

### 3.6. Combination of HS201-PDT with Anti-PD-L1 Antibody Enhanced Antitumor Effect in HER2d16 Model

Although HS201-PDT reduced the incidence of advanced BCs in MMTV-neu mice, the effect was not strong enough to completely block the emergence of advanced tumors. We previously demonstrated the induction of systemic antitumor immunity by HS201-PDT, which caused an abscopal effect on the distant tumors in murine BC implantation models [13]. Moreover, we found that the combination of HS201-PDT treatment with PD-1/PD-L1 blockade (anti-PD-L1 antibody) further enhanced antitumor immunity and the abscopal effect, which lead to greater survival benefits. Therefore, we tested whether an ICI, anti-PD-L1 antibody, would enhance the preventive effect of HS201-PDT on the development of invasive tumors due to induced antitumor immunity. For this purpose, we chose to utilize another HER2-oncogene-driven inducible BC model, HER2d16 mice [31,32]. In this inducible transgenic mouse model, expression of HER2d16 drives rapid tumor onset with a latency of 28 days after initiation of doxycycline (DOX) in the diet. Therefore, we applied HS201-PDT on day 14 after the initiation of a DOX diet and administered anti-PD-L1 antibodies biweekly for 2 weeks, as shown in Figure 5A. Although non-significant, the combination treatment showed a trend of improved DFS compared to the no-treatment control (*p* = 0.084) (Figure 5B). HS201-PDT monotherapy did not show a benefit in DFS in this model. Median DFS was 46, 52 and 68 days for no-treatment, HS201-PDT monotherapy and the combination therapy, respectively. Importantly, the ratio of mice that developed multiple breast tumors was smaller in the combination treatment group (41.7%) compared to the no-treatment control group (71.4%). The ratio of mice that did not develop tumors by 150 days after initiation of the DOX diet was higher in the combination group (33.3%) compared to the control group (14.3%) (Figure 5C). Although these differences were not statistically significant, the results may suggest that the combination treatment tended to delay the progression of DCIS to invasive BC and reduced the number of invasive BCs developing in individual mice.

In our previous studies with human BC xenograft models, we demonstrated the enhanced antitumor efficacy of repetitive HS201-PDT compared to a single treatment [12]. To further improve the efficacy of the combined HS201-PDT and anti-PD-L1 treatment for the prevention of DCIS progression, we repeated HS201-PDT once a week for 4 weeks while administering anti-PD-L1 weekly (Figure 5D). With this repetitive HS201-PDT combined with anti-PD-L1 treatment, the DFS of treated mice significantly improved compared to that of the control group (*p* = 0.0319), with a median DFS of 90 days (Figure 5E). In addition, the ratio of mice that developed multiple breast tumors further decreased in the repetitive combination group (33.3%) compared to the no-treatment control group (71.4%) (Figure 5F).

## 4. Discussion

Current treatment of DCIS with surgery with/without radiotherapy has a high rate of preventing invasive BC; however, overtreatment may occur because many DCIS cases are indolent and may not progress to invasive BC [1,33]. A meta-analysis of unrecognized DCIS showed that it progressed to invasive BC in only 22% of women [34]. A retrospective Surveillance, Epidemiology, and End Results (SEER) study demonstrated that patients with low-grade DCIS had the same overall survival and BC-specific survival rates with or without surgery [33]. A review of autopsy studies reported a median prevalence of DCIS of 8.9%, and a median prevalence of previously undiagnosed invasive BC of 1.3%, suggesting the presence of a large reservoir of undetected, clinically insignificant DCIS in the population [35]. However, there are no reliable biomarkers to predict which DCIS will become invasive BC [6]. To avoid morbidity of surgery, a number of ablative strategies have been developed, including radiofrequency ablation, microwave ablation, cryoablation, high-intensity focused ultrasound and photodynamic ablation [36,37]. Banerjee et al. [38] demonstrated in a small cohort of patients with invasive BC that PDT using VP induced necrosis of breast tumor tissues with no adverse effects.

As a novel treatment strategy for cancers, we developed a Hsp90-targeting PS, HS201, and demonstrated the antitumor efficacy of HS201-PDT against human BC xenografts regardless of their molecular subtypes [12]. Hsp90 is known to be upregulated in many advanced cancers [39,40], including BC, and as we previously demonstrated, BC cells have Hsp90 expression on the cell surface, meaning they will bind HS201 and internalize it into the cytoplasm of cancer cells. As one of the murine spontaneous BC models, we used MMTV-neu mice and demonstrated increased uptake of HS201 by spontaneously inducing invasive breast tumors using body surface imaging, and confirmed HS201 uptake by individual tumor cells via flow cytometry analysis of enzymatically isolated tumor cells. Normal mammary epithelial cells did not take up HS201 [12].

Diehl et al. demonstrated the significantly stronger expression of Hsp90 in the cytoplasm of DCIS cells compared to the normal mammary epithelium by IHC of tissue arrays [17]. In addition, proteomics analysis of human DCIS reported by Wulfkuhle et al. demonstrated that Hsp90 protein was increased in DCIS 26.1-fold relative to matched normal ductal/lobular cells [20]. In our fluorescence microscopy analysis, MCF10DCIS.com cells were shown to have cell surface expression of Hsp90. Therefore, we hypothesized that Hsp90-targeting PDT will exert its cytotoxic effect on DCIS, as was observed in invasive BC, and thus will inhibit its progression to invasive BC. Using the human DCIS cells SUM225CWN and MCF10DCIS.com, we demonstrated that HS201-PDT can inhibit the growth of DCIS tumors implanted into the mammary fat pad of SCID-beige mice. Moreover, in the mammary intraductal implantation model of MCF10DCIS.com cells, HS201-PDT significantly suppressed the tumor growth inside the mammary ducts, lowered the incidence of tumor development to 50% or less and improved survival of mice.

To further recapitulate clinical DCIS cases and to elucidate the involvement of antitumor immunity in the efficacy of HS201-PDT, we next tested spontaneous breast tumors known to develop through a DCIS phase in immunocompetent transgenic mice [30]. We found that applying HS201-PDT to the mammary glands of 5-month-old female MMTV-neu mice delayed the emergence of advanced breast tumors, and thus the treatment significantly improved the disease-free survival. Furthermore, in a more stringent model of spontaneous BC (HER2d16 Tg mice), we observed a greater prevention of DCIS transition to invasive BC following the addition of anti-PD-L1 antibodies to HS201-PDT. This preventative effect was further enhanced by the repetitive administration of HS201-PDT. We hypothesize that HS201-PDT releases tumor antigens, which could serve to induce antitumor immunity. These data support the antitumor immune effect of HS201-PDT against DCIS, which is enhanced by anti-PD-L1 antibodies. Although we have previously observed this induction of antitumor immunity in invasive BC models [13], this is the first observation of the induction of antitumor immunity against DCIS by PDT. While other ablative strategies have reported induction of antitumor immunity, the advantage of PDT is that it is not limited by the maximum exposure dose as radiotherapy is, and thus can be applied multiple times to the targeted area. Further, we achieved this antitumor activity with lower laser energy than in our invasive BC models (30 J/cm^2^ vs. 120 J/cm^2^).

## 5. Conclusions

We have demonstrated that preventive HS201-PDT for DCIS delays the emergence and reduces the incidence of invasive BC in a spontaneous BC model. In a second spontaneous model, the combination of HS201-PDT and anti-PD-L1 antibodies improved tumor-free survival, and repetitive HS201-PDT combined with anti-PD-L1 further prolonged survival time. As a non-invasive treatment for DCIS, HS201-PDT is a promising strategy that could reduce the incidence of invasive BC. Based on our previous findings from a preclinical study on the efficacy of HS201-PDT against advanced BC, we initiated a clinical trial of HS201 (MCT03906643) to first confirm its safety and uptake by malignant breast tumors, and thus GMP-grade HS201 had already been produced. Based on these preliminary data, we aim to optimize the combination strategy schedule (PDT + ICI) for clinical applications.

## Figures and Tables

**Figure 1 cancers-14-05762-f001:**
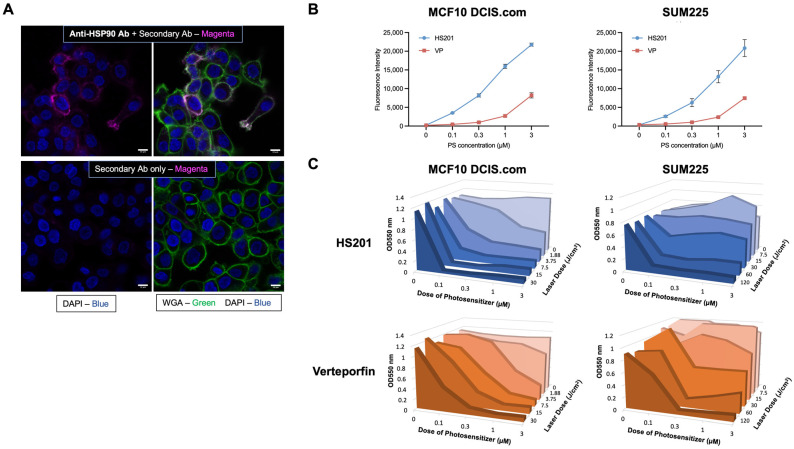
In vitro uptake of HS201 and cytotoxicity of HS201-PDT against human DCIS cell lines. (**A**) Cell surface expression of Hsp90 on DCIS cells. MCF10DCIS.com cells cultured on glass-bottom dishes were incubated with or without anti-Hsp90 antibody (clone AC88, Abcam) for 30 min on ice, then fixed with 10% neutral buffered formalin. Then, cells were incubated with AF568-conjugated goat anti-mouse IgG, followed by WGA AF488 conjugate membrane staining dye and DAPI. After being washed with PBS, cells were observed using a confocal microscope. Upper images: with anti-Hsp90 antibody. Bottom images: secondary antibody alone. Magenta: Hsp90, Green: cell membrane, Blue: DAPI. Scale bars: 10 µm. (**B**) In vitro uptake of photosensitizers HS201 and parental VP by human DCIS cell lines. MCF10DCIS.com and SUM225CWN cells were cultured in 96-well plates, and VP or HS201 (0.03–3 μM, respectively) was added, incubated for 30 min at 37 °C and then washed with PBS. The signal intensity of each well was measured at 700 nm by LI-COR Odyssey imager. (**C**) Cytotoxicity of in vitro PDT using VP (VP-PDT) and HS201 (HS201-PDT) determined by MTT assay. DCIS cells in a 96-well plate were co-incubated with PS and washed as described in Figure 1B. A laser with a wavelength of 690 nm was applied at various doses (1.88–120 J/cm^2^) to each well, and cells were incubated overnight prior to the analysis. An MTT assay was performed, and the optical density values at 550 nm (650 nm subtracted) are shown.

**Figure 2 cancers-14-05762-f002:**
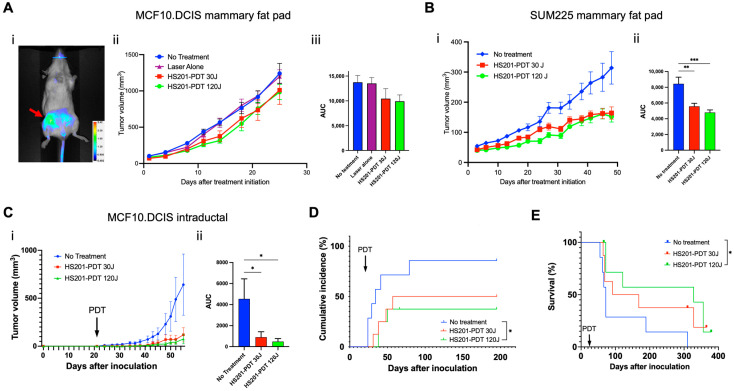
Antitumor efficacy of in vivo HS201-PDT against human DCIS cell lines. Antitumor efficacy of HS201-PDT against MCF10DCIS.com cells in mammary fat pad implantation model. MCF10DCIS.com cells and SUM225CWN cells (1 × 10^6^ cells/mouse) were inoculated into the 4th mammary fat pad of female SCID-beige mice. After the tumor size reached 5 mm in diameter, PDT was performed. (**A**) Six hours after tail vein injection of HS201, nIR images of the body surface were taken for the MCF10DCIS.com-implanted mouse (**i**). A 690 nm laser was used to irradiate the 4th mammary gland at the doses of 30 and 120 J/cm^2^. Tumor size was monitored by caliper measurement twice a week, and average tumor volumes for each group are shown (**ii**). N = 10 mice for both no-treatment and HS201-PDT 120J groups, and n = 9 mice for both laser alone and HS201-PDT 30J groups. Right panel (**iii**) shows the comparison of the area under the curve (AUC) of tumor growth curves for no-treatment control, laser alone (120 J/cm^2^) and HS201-PDT groups with 30 or 120 J/cm^2^ laser. (**B**) Growth of SUM225CWN tumors after HS201-PDT was monitored, and average tumor volumes are shown (**i**). N = 10 mice for all the groups. AUC of tumor growth curves was compared (**ii**). (**C**) In intraductal implantation model, twenty-one days after intraductal injection of MCF10DCIS.com cells to the 4th mammary gland (arrow), mice received HS201-PDT as described above. N = 7 mice for no-treatment group, and n = 8 mice for both HS201-PDT 30J and HS201-PDT 120J groups. Average tumor volumes are shown (**i**). AUC of tumor growth curves was compared (**ii**). (**D**) Cumulative incidences of tumors after intraductal tumor implantation are shown for each treatment group. PDT was performed on day 21 (arrow). (**E**) Kaplan–Meier survival curves of mouse overall survival are shown. Log-rank test was performed for the statistical analysis. *p* value: * *p* < 0.05, ** *p* < 0.01, *** *p* < 0.001.

**Figure 3 cancers-14-05762-f003:**
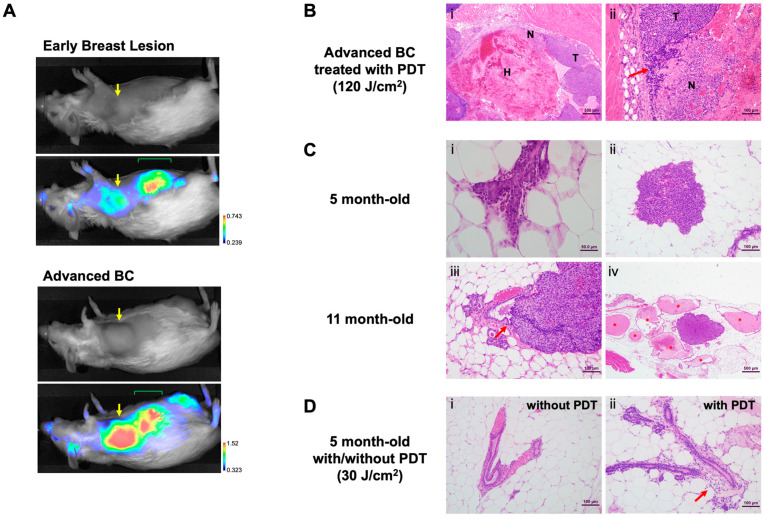
Development of DCIS lesions in MMTV-neu mice. (**A**) An MMTV-neu mouse at the age of 7 months with DCIS lesion in the right #3 mammary gland and another mouse at the age of 10 months with an advanced breast tumor were administered HS201 (25 nmol/mouse) via tail vein injection, and 6 h later, nIR images were taken using LI-COR Pearl imager. Yellow arrows indicate the locations of the breast lesions. At this stage of imaging, HS201 was mostly secreted into bile, then into the intestine and detected as nIR signals in the abdomen (green mark). (**B**) Pathological analyses of HS201-PDT-treated breast cancers in MMTV-neu mice. (**i**) Low magnification photomicrograph of MMTV-neu mammary carcinoma treated with HS201-PDT showing extensive hemorrhage (H) and necrosis (N). H&E bar = 500 μm. (**ii**) High-magnification photomicrograph of mass (i) showing area of viable neoplasm (T), necrosis (N) and infiltrate of lymphoid cells (arrow). H&E bar = 100 μm. (**C**) Pathological analyses of spontaneous breast lesions in MMTV-neu mice. (**i**) Photomicrograph of mouse mammary duct showing very early dysplasia manifesting as proliferation of ductal epithelial cells with cellular atypia. H&E bar = 50 μm. (**ii**) Early MIN lesion showing typical cytologic features of MMTV-neu mammary tumors. H&E bar = 100 μm. (**iii**) Large MIN lesion arising from epithelial cells of duct wall (arrow). H&E bar = 100 μm. (**iv**) MIN lesion surrounded by duct ectasia (*) suggesting ductal origin of lesion. H&E bar = 500 μm. (**D**) Effect of preventive HS201-PDT on mammary glands in 5-month-old MMTV-neu mice. (**i**) Normal mammary duct in adipose tissue, in contrast with (**ii**) a duct from a PDT-treated mouse demonstrating periductal leukocytic infiltration (arrow). H&E bar = 100 μm.

**Figure 4 cancers-14-05762-f004:**
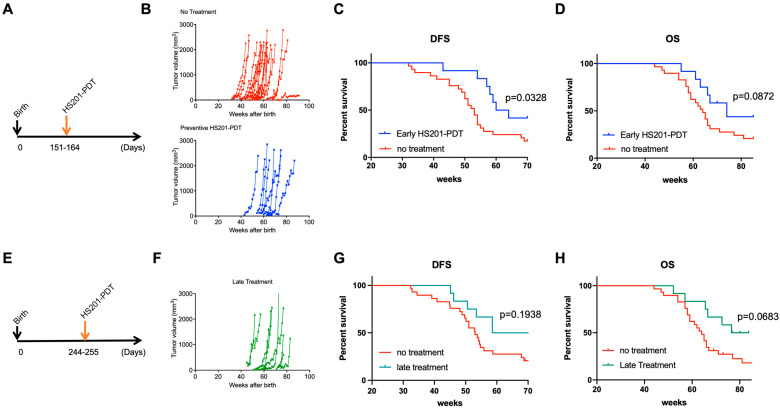
Survival benefits of preventive HS201-PDT for MMTV-neu mice. (**A**) Preventive HS201-PDT was conducted for 5-month-old (151–164 days after birth) female MMTV-neu mice. Six hours after intravenous administration of HS201 (25 nmol/mouse), a 30 J/cm^2^ laser dose was used to irradiate each of 7 irradiation areas. Tumor growth was monitored every week until humane endpoints or till the age of 85 weeks. Control mice in no-treatment group were monitored for tumor emergence and tumor growth without any treatment. N = 29 for no-treatment group and n = 12 for early HS201-PDT group. (**B**) Tumor growth for individual mice is plotted. (**C**) Kaplan–Meier curves of disease-free survival are shown for HS201-PDT early treatment group and no-treatment control group. (**D**) Kaplan–Meier curves of overall survival are shown for HS201-PDT early treatment group and no-treatment control group. (**E**) Preventive HS201-PDT was conducted for 8-month-old (244–255 days after birth) female MMTV-neu mice. HS201-PDT was performed, and tumor growth was monitored until humane endpoints or till the age of 85 weeks, as described in (**A**). N = 29 for no-treatment group and n = 12 for late HS201-PDT group. (**F**) Tumor growth for individual mice is plotted. (**G**) Kaplan–Meier curves of disease-free survival are shown for HS201-PDT late treatment group and no-treatment control group. The same survival curve for no-treatment control group is shown for comparison purposes. (**H**) Kaplan–Meier curves of overall survival are shown for HS201-PDT late treatment group and no-treatment control group.

**Figure 5 cancers-14-05762-f005:**
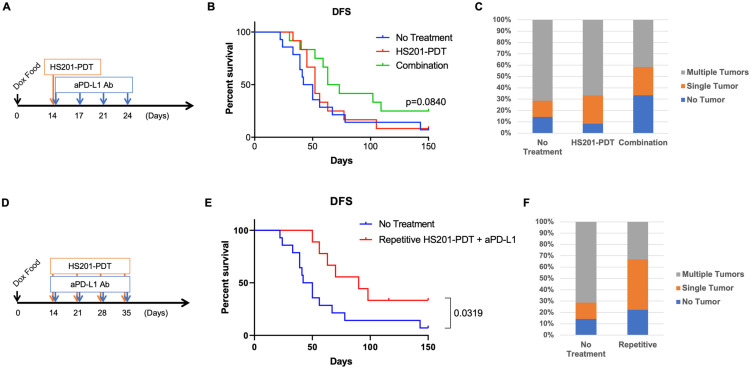
Combination strategies of preventive HS201-PDT and anti-PD-L1 antibody. (**A**) Schedule of anti-PD-L1 administration combined with a single HS201-PDT treatment. Mice were given doxycycline diet starting on day 0, and HS201-PDT was given on day 14. Starting on the same day, anti-PD-L1 antibody (250 µg/injection) was intraperitoneally administered to mice twice a week for 2 weeks. N = 14 mice for no-treatment group, n = 12 for both HS201-PDT monotherapy group and the combination therapy group. (**B**) Disease-free survivals of HER2d16 Tg mice are shown. X axis shows the days after the initiation of doxycycline diet. Mice were monitored until day 150 after the initiation of doxycycline diet. Pairwise comparison between the combination group and no-treatment control group showed *p* value= 0.084. (**C**) The graph shows the incidence of mice with no tumor, a single tumor and multiple tumors, based on treatments. (**D**) Schedule of anti-PD-L1 administration combined with weekly HS201-PDT treatments. Mice were given doxycycline diet starting on day 0, and HS201-PDT was given weekly on days 14, 21, 28 and 35. On the same days PDT was conducted, anti-PD-L1 antibody (250 µg/injection) was intraperitoneally administered to mice. N = 14 mice for no-treatment group, and n = 9 for repetitive combination group. (**E**) Disease-free survivals of HER2d16 Tg mice treated with repetitive combination of HS201-PDT and anti-PD-L1 are shown. Mice were monitored until day 150 after the initiation of doxycycline diet. (**F**) The graph shows the incidence of mice with no tumor, a single tumor and multiple tumors, based on treatments. The same control group data as those in (**C**) were used for the comparison.

## Data Availability

The data that support the findings in this study are available upon reasonable request from the corresponding author.

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
