# Peer review of "A Non-Invasive Deep Photoablation Technique to Inhibit DCIS Progression and Induce Antitumor Immunity"

_cancers, 2022, doi:10.3390/cancers14235762_

Round 1

Reviewer 1 Report

In this study, the authors investigated the effects of photoablation technique on DCIS progression. They found that preventive HS201-PDT for DCIS delays the emergence and reduces the incidence of invasive BC. The combination of HS201-PDT and anti-PD-L1 antibody improves tumor free survival, and repetitive HS201-PDT combined with anti-PD-L1 further prolongs survival time. The observation in this study is interesting, but there are several concerns that should be addressed.

Although many interesting observations in this study, it seems there is a lack of further mechanism exploration.

The reviewer would like to know why the authors employed the laser energy condition of 30 J/cm2 and 120 J/cm2, they should describe the reasons in the manuscript.

Verteporfin was used in this study as a photosensitizer for photodynamic therapy. Verteporfin is also a YAP inhibitor, which can destroy the interaction of YAP and TEAD, and can induce cell apoptosis. How about the effect of Verteporfin on DCIS cells? Do the authors have this piece of information?

As the authors mentioned that the differences of some results in this study were not statistically significant, thus it maybe need to do more experiments to confirm the conclusion.

Author Response

Reviewer 1

Comments and Suggestions for Authors

In this study, the authors investigated the effects of photoablation technique on DCIS progression. They found that preventive HS201-PDT for DCIS delays the emergence and reduces the incidence of invasive BC. The combination of HS201-PDT and anti-PD-L1 antibody improves tumor free survival, and repetitive HS201-PDT combined with anti-PD-L1 further prolongs survival time. The observation in this study is interesting, but there are several concerns that should be addressed.

Although many interesting observations in this study, it seems there is a lack of further mechanism exploration. 

RESPONSE: In this study, we mainly focused on survival benefit of the treatment, and thus mice were observed until they came to humane endpoints. For the mechanism exploration, especially the involvement of antitumor immunity, we believe the assessment needs to be performed at earlier time points, but not at the humane endpoints. Therefore, we plan to conduct mechanism exploration study in our next research, by euthanizing mice at determined time points (such as, 7 days or 14 days after PDT) to assess the changes of immune cells in PDT-treated mammary glands using single cell RNA-sequencing and multispectral microscopy. We anticipate that this study will take at least a year, including mouse preparation, observation for tumor growth, conduct of assays, and data analysis.

The reviewer would like to know why the authors employed the laser energy condition of 30 J/cm2 and 120 J/cm2, they should describe the reasons in the manuscript.

RESPONSE: The laser doses were determined from our previous studies (Commun Biol (ref#12), JITC (ref#13)), in which we performed HS201-PDT for human BC xenografts in SCID mice and murine BC tumors in immunocompetent mice. Based on these studies and preliminary experiments conducted to determine optimal laser doses, we found 30-120 J/cm2 were the optimal laser doses to significantly suppress tumor growth without heating or burning of tissues. In the current study, however, our preliminary experiment to determine the laser dose for preventive PDT showed mild liver damage in some of the mice that received 120 J/cm2 of laser light to the right 2nd-3rd mammary glands. No liver damage was observed when 30 J/cm2 of laser light was applied to the area. As shown in Figure 2, both 30 J/cm2 and 120 J/cm2 of laser showed comparable antitumor efficacies in human DCIS models. Based on these findings, we chose the safer dose (30 J/cm2) of laser light for the preventive PDT in transgenic mouse models to avoid adverse events. We added a brief explanation of the reason for this laser dose in the Materials and Methods section (page 5, lines 199-200, 212-213).

Verteporfin was used in this study as a photosensitizer for photodynamic therapy. Verteporfin is also a YAP inhibitor, which can destroy the interaction of YAP and TEAD, and can induce cell apoptosis. How about the effect of Verteporfin on DCIS cells? Do the authors have this piece of information?

RESPONSE: We did not test the effect of verteporfin for YAP or Hippo pathway in DCIS cells, but in our cytotoxicity assay (MTT assay) performed after in vitro PDT to the DCIS cells, we did not observe significant cell killing if cells were exposed to verteporfin without laser light irradiation (Figure 1C, data for laser dose 0). OD550 nm values for verteporfin alone conditions (without laser irradiation) are shown in the table below.

OD550 nm value

                                             Verteporfin (µM)

0

0.1

0.3

1

3

MCF10DCIS.com

1.153 +/- 0.089

1.097 +/- 0.093

1.078 +/- 0.062

1.080 +/- 0.079

1.106 +/- 0.081

SUM225

1.071 +/- 0.161

1.344 +/- 0.040

1.164 +/- 0.032

1.164 +/- 0.028

1.038 +/- 0.028

In the published literature, suppression of tumor cell growth by verteporfin as a YAP inhibitor was observed when relatively higher concentrations (3 or 4-16 µM) of verteporfin were used in the assays (Wang et al., Am J Cancer Res 2015, Wei et al., BMC Cancer 2020, Guimei et al., BCTT 2020), but our in vitro PDT was tested using lower concentrations (0.1, 0.3, 1 and 3 µM) of verteporfin. Therefore, we assume the cytotoxicity induced by verteporfin as a YAP inhibitor was negligible or minimal at the low concentrations in the current study.

As the authors mentioned that the differences of some results in this study were not statistically significant, thus it maybe need to do more experiments to confirm the conclusion.

RESPONSE: We plan to conduct additional experiments 1) to determine antitumor efficacy and survival benefit and 2) to explore the mechanisms of action in which we will collect breast tissue/tumor samples from mice by euthanizing them before humane endpoints. However, because it takes over 80 weeks (almost 1.5 year) to assess the emergence and the growth of spontaneous BC in MMTV-neu mice, we would need to perform these experiments as a part of a future study. We will also conduct more experiments using HER2delta16 Tg mice to evaluate the efficacy of the combination therapy, but it will take at least 6-8 months to prepare a large enough number of mice to perform these treatments and analyze the data. Thus, we propose to conduct these experiments of combination therapy as a part of our next studies. 

Reviewer 2 Report

This is very elegantly prepared work, well described and illustrated, presenting very interesting experiments. I would like to suggest only: could the Authors show in supplementary material include a photo presenting irradiation of the mouse using a phantom/model mouse?

Author Response

Comments and Suggestions for Authors

This is very elegantly prepared work, well described and illustrated, presenting very interesting experiments. I would like to suggest only: could the Authors show in supplementary material include a photo presenting irradiation of the mouse using a phantom/model mouse?

RESPONSE: We added a photo showing the laser irradiation system used in this study in Supplementary Figure S1.

Round 2

Reviewer 1 Report

The authors have made efforts to revise the manuscript. Their revisions are appropriate and the response to my comments is satisfactory. To this reviewer, the revised manuscript is now suited for publication in Cancers.